# *Hanensula anomala* isolated from the Berkeley Pit, Butte, MT, is a metal-specific extremophile

Kyle Roessler,[1] Mariah C. Friedlander,[2] Marthe Y. VanSickle,[3] Christina L. Rush[1]

**ABSTRACT** A yeast-like extremophile organism, *Hansenula anomala*, has been isolated from the superfund site the Berkeley Pit Lake in Butte, Montana. Studies demonstrate *H. anomala* growth in some of the known Berkeley Pit Lake solutes. Microbial growth dynamics under controlled conditions were compared of *H. anomala* for multiple metal concentrations. Each solute/metal was tested separately at previously reported concentrations on the geochemistry of the Berkeley Pit lake in the first 0.2 m in spring (pH 2.5). *H. anomala* grew well with sulfur (S), $MgSO_4$, $CaSO_4$, potassium chloride (KCl), and $NaSO_4$ and was inhibited with $FeSO_4$, $MnSO_4$, $CuSO_4$, $AlSO_4$, or $ZnSO_4$. With the addition of elemental S, growth was observed for $FeSO_4$ indicating minimal growth rescue. PCR amplification of genomic DNA from the organism using known ribosomal primers indicates the strain to be ATCC8168 (CBS 5759). From this data, it can be concluded that *H. anomala* ATCC8168 from the Berkeley Pit is an extremophile that exhibits metal-specific growth.

**IMPORTANCE** Laboratory growth studies of a strain of *Hansenula anomala* from the Berkeley Pit have found the organism to be metal specific indicating some unique metabolism possibilities. These studies show that this strain is metal-dependent and provides information about the adaptable tolerance of organisms in superfund sites as well as giving a basis for future bioremediation development utilizing *H. anomala*.

**KEYWORDS** bioremediation, yeasts, water quality, metabolism, identification

Extremophile microbes have been discovered in many different environments previously thought to be devoid of life, or unable to support thriving life (1, 2). Extremophiles are microbes that have the ability to grow in environments that exceed the boundaries of most life including very high or low pH, high salt, very high heat, or very cold environments (3, 4). These microbes are found living on iceburgs, in famous hot springs, in the dead sea, and in mining waste sites. Now, more than ever, the biotechnological and pharmaceutical applications from extremophile byproducts and the microbes themselves are needed during this time of climate change and the discovery for alternative energy (5). They are important for rising discovery in healthcare needs for growing world populations (6, 7). Newly discovered and/or researched extremophiles are also key for novel bioremediation efforts and bioaugmentation (8).

*Hansenula anomala*, (*Pichia anomala*, *Wickerhamomyces anomalus*), is a yeast-like organism that has been isolated from a number of harsh environments including anaerobic feed storage units, the intestine of *Anopheles stephensi* (a malaria vector), and low pH metal toxic waste water from a textile plant in Argentina (9–11). The taxonomy of *H. anomala* varies with some phylogenetic studies placing the organism to species within the genus *Hansenula* (12). Another genus, *Wickerhamonyces*, is also found in the literature from separate taxonomic grouping; however, for simplicity, this research report will use *H. anomala* as the organism's name (12, 13).

Address correspondence to Kyle Roessler, kyle_roessler@skc.edu, Christina L. Rush, christina_rush@skc.edu, Mariah C. Friedlander, info@energykeepers.org, or Marthe Y. VanSickle, clerkofcourt@missoulacounty.us.

The authors declare no conflict of interest.

**Received** 19 Februuary 2024

10.1128/spectrum.00444-24 **1**

Within the various strains of *H. anomala*, many show antimicrobial activity, and *H. anomala* has been shown to inhibit gram-negative bacteria including *Enterobacteriaceae* species and filamentous fungi including *Aspergillus*, *Botrytis*, and *Penicillium* species (14–17). The mechanism of inhibition may be due to "killer proteins" described to consist of a variety of molecular masses where the variation in size may be attributed to various levels of glycosylation (14, 18, 19). One of the "killer proteins," a novel exo-β-1,3-glucanase, was isolated from the *H. anomala* strain NCYC 434 in 2011 and revealed an activity attributed to the inhibition of fungal colonization on fruit including apples and citrus thus suggesting an attractive possibility for *H. anomala* to act as a biocontrol agent (15, 16, 20, 21).

Phytase activity is an additional characteristic in a number of *H. anomala* strains, from which a phytase gene was sequenced to utilize it as a biotechnological tool in removing phosphorous from animal waste to reduce the overall environmental footprint of farms (17, 22, 23).

*H. anomala* has also demonstrated promise in the bioremediation of high concentration metal-containing wastewater. Specifically, M10, a chromate-resistant strain, was shown to remove toxic chromium ($Cr^{6+}$) in batch cultures using cell-free extracts suggesting the potential for the restoration of left-over waste-mining sites (11).

The geochemistry of the Berkeley Pit includes a high concentration of metals and organic compounds, which vary by depth, increasing dramatically in concentration for metals, specifically iron and sulfate (24–26). Moreover, metal concentrations increase seasonally recording a higher measured concentration in the fall as opposed to the spring. This variation is possibly due to rain and snow melt creating a dilution effect in the spring versus the fall after a hot and dry summer (24–26).

Because of the unique environment, from which our isolate was found, we hypothesized that there would be interesting growth patterns. Here, we show solute and metal-repressive and/or -suppressive growth from the Berkeley Pit Lake isolate, *H. anomala* strain ATCC8168 (CBS 5759) providing experimental data on the adaptable tolerance of this organism in superfund sites.

## MATERIALS AND METHODS

### Liquid media preparation and growth of *H. anomala* under controlled conditions

Liquid media was prepared at a pH of 2.5 using sulfuric acid, 6 g of potato dextrose, and varying combinations of metals. For each media preparation, 6 g of potato dextrose was added to 100 mL of MilliQ water and autoclaved for sterility. Sulfuric acid was added drop-wise to cooled media in a sterile hood for a final pH of 2.5 at room temperature. Metals were dissolved in 100 mL MilliQ water and sterile filtered or autoclaved depending on metal solubility. The metal preparation was then added to room temperature potato dextrose media at a pH of 2.5 with sulfuric acid addition. The final volume of each prepared media was adjusted to 250 mL using MilliQ water. Media was stored at 4°C for no longer than 8 days or used immediately. All of the metals were sulfate salts except for KCl and S to mimic the extremely high concentration of sulfate ($SO_4$) in the Berkeley Pit lake. KCl salt was used due to the common concentration reported for K and Cl in the Berkeley Pit Lake. S was also tested due to the high level of S released in the lake.

*H. anomala* was first cultured onto potato dextrose agar plates with a pH of 2.5 from previously isolated samples generously given from Andrea Stierle Ph.D. of the University of Montana in 2011. The plates were streaked for colony isolation and incubated at 30°C for 72 h. All samples and plates were kept under sterile hood conditions for streaking to prevent contamination.

Single isolated colonies of *H. anomala* were inoculated into the prepared liquid media of potato dextrose media at a pH of 2.5 with sulfuric acid from the potato dextrose agar plate pH 2.5 using a sterile, disposable inoculation loop into 1.5 mL of the media in the absence of metal/solute and incubated at 30°C, overnight at 180 Revolutions Per Minute

(RPM). The purpose of growth in the absence of any metal was to promote confluent growth in liquid culture before inoculation into the metal/solute-containing liquid media for study. The 1.5 mL inoculates were then added to 50 mL of metal-containing media and grown overnight at 30°C at a 180 RPM. All inoculations and *H. anomala* culturing were carried out in a sterile fume hood.

The 50 mL inoculations were incubated for 72 h at 30°C at a 180 RPM and sampled each day at the same time by removing the flasks from the shaking incubator for sampling from a sterile fume hood. A 1 mL sample was measured for growth and confluence using Optical Density (OD) measurements against un-inoculated liquid media.

The OD measurements were taken three times a week, over a 72 h growth period to calculate growth. By 24 h, samples were confluent, and serial dilutions with un-inoculated media were carried out. Dilutions were calculated to reflect cell growth, and for replication, all growth measurements were conducted three times for each metal/solute liquid media. Statistical analysis of growth data were thus performed from triplicate samples. After 72 h sampling and OD measurements, the samples were centrifuged, and pellets were frozen for future protein profile characterization.

## Genomic DNA extraction and PCR analysis

For genomic DNA extraction, *H. anomala* was grown in potato dextrose liquid media at a pH of 2.5 at 30°C for 24 h to confluency in a 1 mL preparation at an RPM of 150. Cells were pelleted via centrifugation and genomic DNA was isolated from *H. anomala* using the QIAGEN Dneasy blood & tissue kit 250 and the accompanying protocol "Purification of total DNA from yeast using the Dneasy blood & tissue kit (DY13 Aug 06)."

PCR amplification was carried out with primers:

ITS3 GCATCGATGAACGCAGC and ITS4 TCCTCCGCTTAATTGATATGC, NS1 GTAGTCATAT GCTTGTCTC and NS-8 TCCGCAGGTTCACCTACGGA, ITS1 TCCGTAGGTGAACCTGCGG and NS-8A  CCTTCCGCAGGTTCACCTACGGAAACC,  NL-1  GCATATCAATAAGCGGAGGA-AAAG and NL-4 GGTCCGTGTTTCAAGACGG, NL-3A GAGACCGATAGCGAA-CAAG and NL-7AR C CGACTTCCATGGCCACCGTCC, NL-E27 GGTAGCCAAT-GCCTCGTCA and NL-11R CCTTGTCC GTACCAGTTCTAAGT, NLG19A GGGA-ACGTGAGCTGGGTTTAGACCG and NL-13R GCGTTA TCGTTTAACAGATGTG-CCG (13, 27, 28) and *Taq* polymerase (1× standard *Taq* reaction buffer pack, New England Biolabs) at standard conditions: for a 25 µL reaction, 2.5 µL 10× standard *Taq* reaction buffer, 0.5 µL 10 mM dNTPs, 1 µL 10 µL FWD primer, 1 µL 10 µL REV primer, ~50 ng genomic DNA (varied from extraction), 0.5 µL *Taq* DNA polymerase, QS to 25 µL with nuclease-free water. Thermocycling conditions were set up for ribosomal and mitochondrial gene amplification from genomic DNA with an initial denaturation step for 1 min at 95°C followed by 35 cycles (30 s at 95°C, 60 s at 48°C, 2 min 68°C), and a final extension 72°C 5 min. PCR amplifications were analyzed via 1% molecular grade agarose with SYBR safe DNA gel stain.

## RESULTS

### PCR strain analysis of *H. anomala* isolated from the Berkeley Pit

PCR analysis was conducted with isolated genomic DNA to determine the strain of *H. anomala* used in the growth study (originally isolated from the Berkeley Pit Lake). Primers were designed from previous data by researchers who have previously sequenced *H. anomala* strains (13, 27, 28). PCR analysis showed amplification of bands for primers

**TABLE 1** Solute concentrations for Berkely Pit Lake, Spring 2003 by Pellecori 2005 at a depth of 0.2 m and liquid media preparation solute concentrations[a]

|  | Al | Ca | Cu | Fe III | Mg | Mn | K | Na | Zn | SO4 |
|---|---|---|---|---|---|---|---|---|---|---|
| Berkeley pit lake | 134 | 420 | 64.8 | 263 | 302 | 161 | 14 | 70 | 275 | 4,530 |
|  | AlSO$_4$ | CaSO$_4$ | CuSO$_4$ | FeSO$_4$ | MgSO$_4$ | MnSO$_4$ | KCl | NaSO$_4$ | ZnSO$_4$ | S |
| Liquid media | 134 | 420 | 64 | 263 | 300 | 161 | 19 | 70 | 275 | 4.53 |

[a]All concentrations are in mg/L.

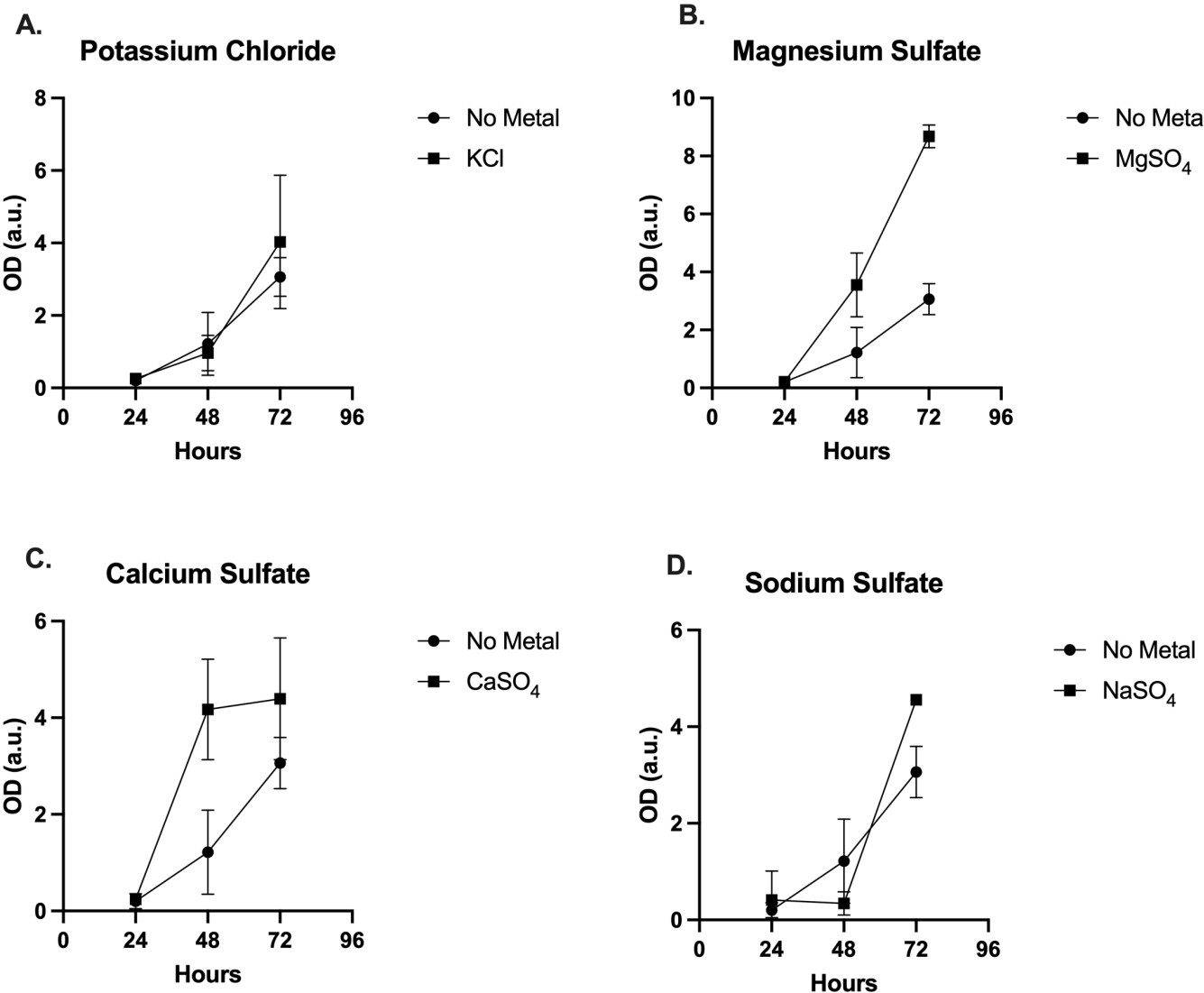

**FIG 1** Base media added solutes/metals KCl, MgSO$_4$, CaSO$_4$, and NaSO$_4$. (A) *H. anomala* grew at nearly the same rate as no added metal in the presence of KCl. (B) Growth of the organism was slowed by approximately half with the addition of MgSO$_4$. (C) CaSO$_4$ was within the error rate for matching growth with no added metal. (D) NaSO$_4$ appeared to increase *H. anomala* growth in relationship to no added solute (metal).

described in the methods section and indicated the strain to be ATCC8168 (CBS 5759) (Fig. S1).

### *H. anomala* minimal growth requirements

Initial growth studies in minimal media were conducted with varying additives including tryptophan, lysine, arginine, valine, proline, glucose, skim milk, lactose, fructose, and cellulose (Supplementary data, Table 1). A summary of these initial results is that *H. anomala* ATCC8168 could grow in less acidic pH up to 6.5. The organism does not appear to have any essential amino acids as it has average growth in minimal base media; however, there is a suppressed growth in the presence of skim milk (powdered), and when lactose or cellulose was added. M9 salts with added metals and glucose also show no growth. Finally, the organism was shown to be catalase positive and urease positive (Supplementary data, Table 1).

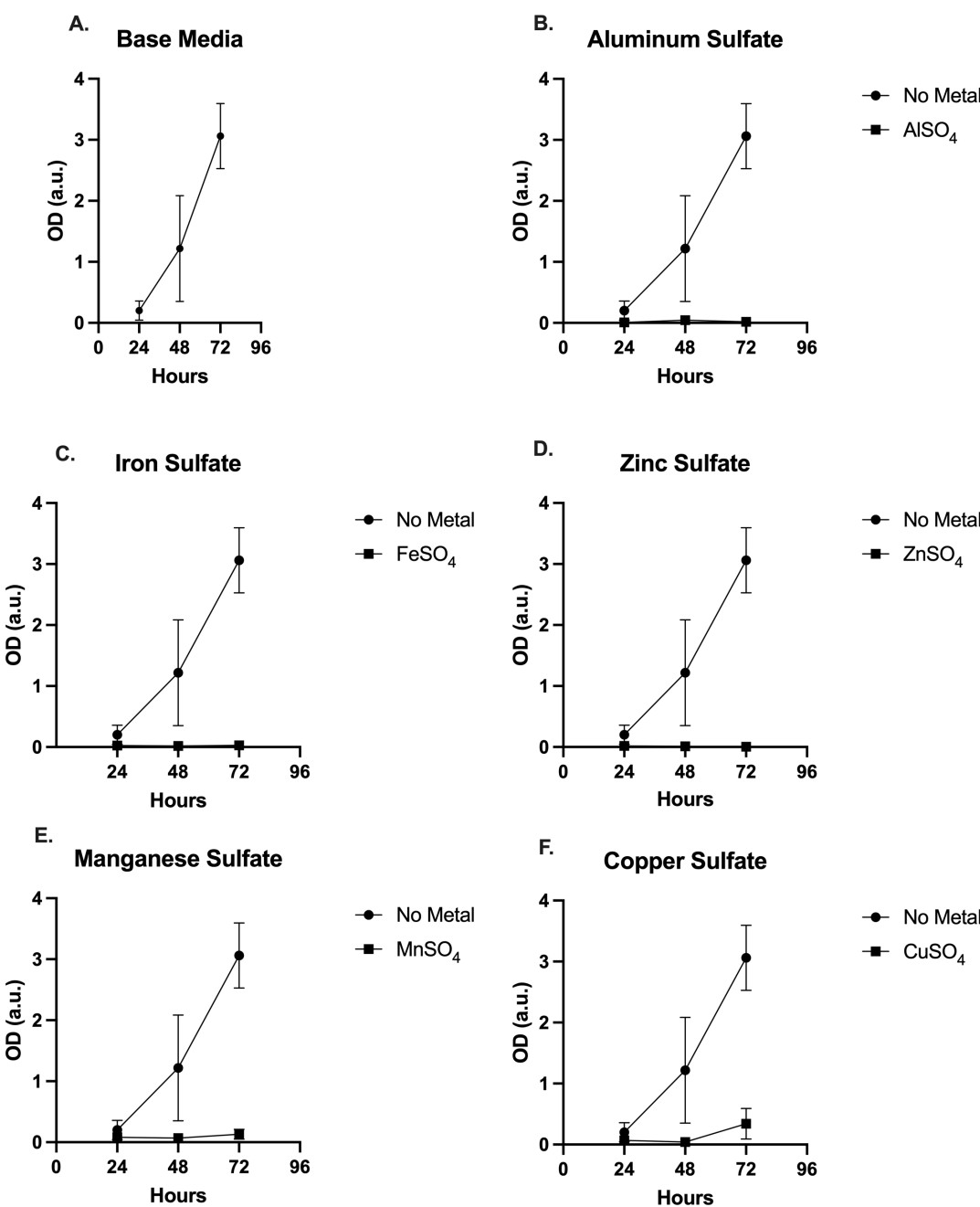

**FIG 2** Base media added solutes/metals AlSO$_4$, FeSO$_4$, ZnSO$_4$, MnSO$_4$, and CuSO$_4$. (A) *H. anomala* growth in base media. (B) No observed growth with added AlSO$_4$. (C) *H. anomala* exhibited no growth when FeSO$_4$ was added to base media. (D) No measurable growth was recorded with the addition of ZnSO$_4$ to base media. (E) Very little measurable growth was observed with the addition of MnSO$_4$ to base media at 72 h post inoculation. (F) Some growth was recorded for CuSO$_4$ additive media at 72 h post inoculation.

## Metal-specific growth studies

*H. anomala* ATCC8168 (CBS 5759) growth with specific metals/solutes was measured in a liquid media of nutrient potato dextrose media at a pH of 2.5 with sulfuric acid and showed significant differences in growth patterns for different metal/solute conditions over the course of 24–72 h growth periods. The laboratory metal/solute concentrations used were consistent with the solute concentrations measured by Pellicori *et al.* (25) for the Berkeley Pit Lake in spring at a depth of 0.2 m (Table 1). Fe(II) was not measured

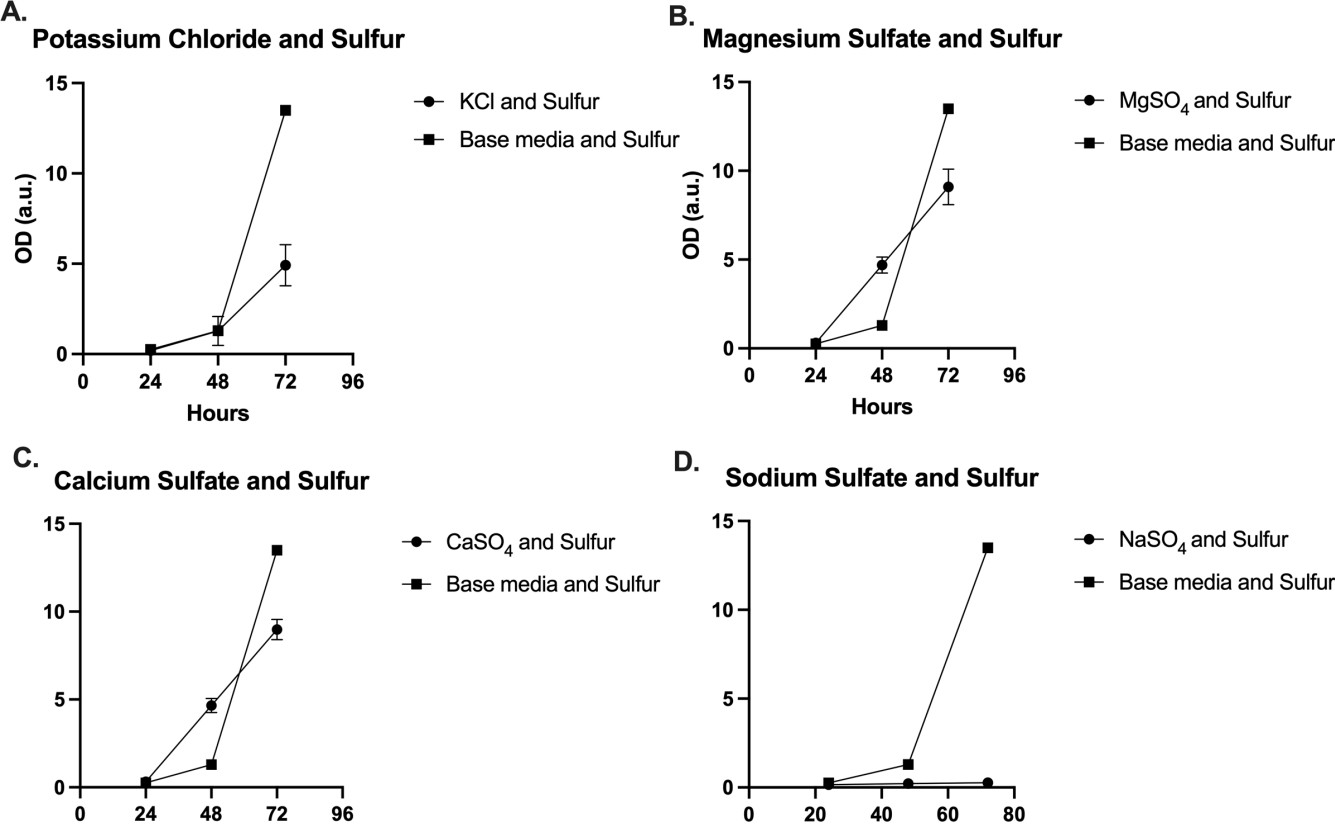

**FIG 3** Base media with elemental S and added solutes/metals KCl, MgSO$_4$, CaSO$_4$, and NaSO$_4$. (A) *H. anomala* had increased growth with KCl and elemental S than without S. (B and C) Organism growth was nearly as high as base media and S with added MgSO$_4$ and CaSO$_4$. (D) Growth with NaSO$_4$ and elemental S was suppressed.

because its concentration at 0.2 m depth is negligible compared with Fe(III). The pH of the Berkeley Pit is acidic with a fifteen-year mean pH of 2.63 (26).

*H. anomala* does not require metal or salts for growth as it was shown to grow well in base media and with added elemental S at a concentration of 4530 mg/L (25), suggesting the organism has most likely evolved to utilize S in metabolism pathways rather than simply tolerating it (Table 1). Additionally, the organism also showed a significant growth in a base media of potato dextrose media at a pH of 2.5 with sulfuric acid. Individual metals were added to the base media (Fig. 1) (A) magnesium (302 mg/L), (B) calcium (420 mg/L) (C) potassium (14 mg/L), and (D) iron (263 mg/L). Each of these individually added metals showed significant growth.

Interestingly, although all the tested metals and solutes were derived from identified contaminants of the Berkeley Pit, five metals suppressed growth when added to the base media. Figure 2 shows negligible growth in base media with (B) FeSO$_4$, (C) MnSO$_4$, (D) CuSO$_4$, (E) AlSO$_4$, or (F) ZnSO$_4$ (25). To examine whether the metals such as iron, manganese, copper, aluminum, or zinc were actually killing the organism or simply inhibiting growth, inoculates from the 72 h liquid media were streaked out on solid base media and incubated for 24 h at 30°C. Robust colony growth was observed indicating an inhibition of growth in the metal-containing media, not cell death (results not shown).

To further probe the nature of the inhibition, liquid base media with the suppressive metals was inoculated and then provided with additional elemental S. The hypothesis of the inhibition process was that added elemental S could reverse the suppressive effects of the metals in question. As a control, the initial metals/solutes that resulted in a significant organism growth in base media were tested with elemental S (Fig. 3). Surprisingly, NaSO$_4$ in base media with elemental S had negligible growth even over

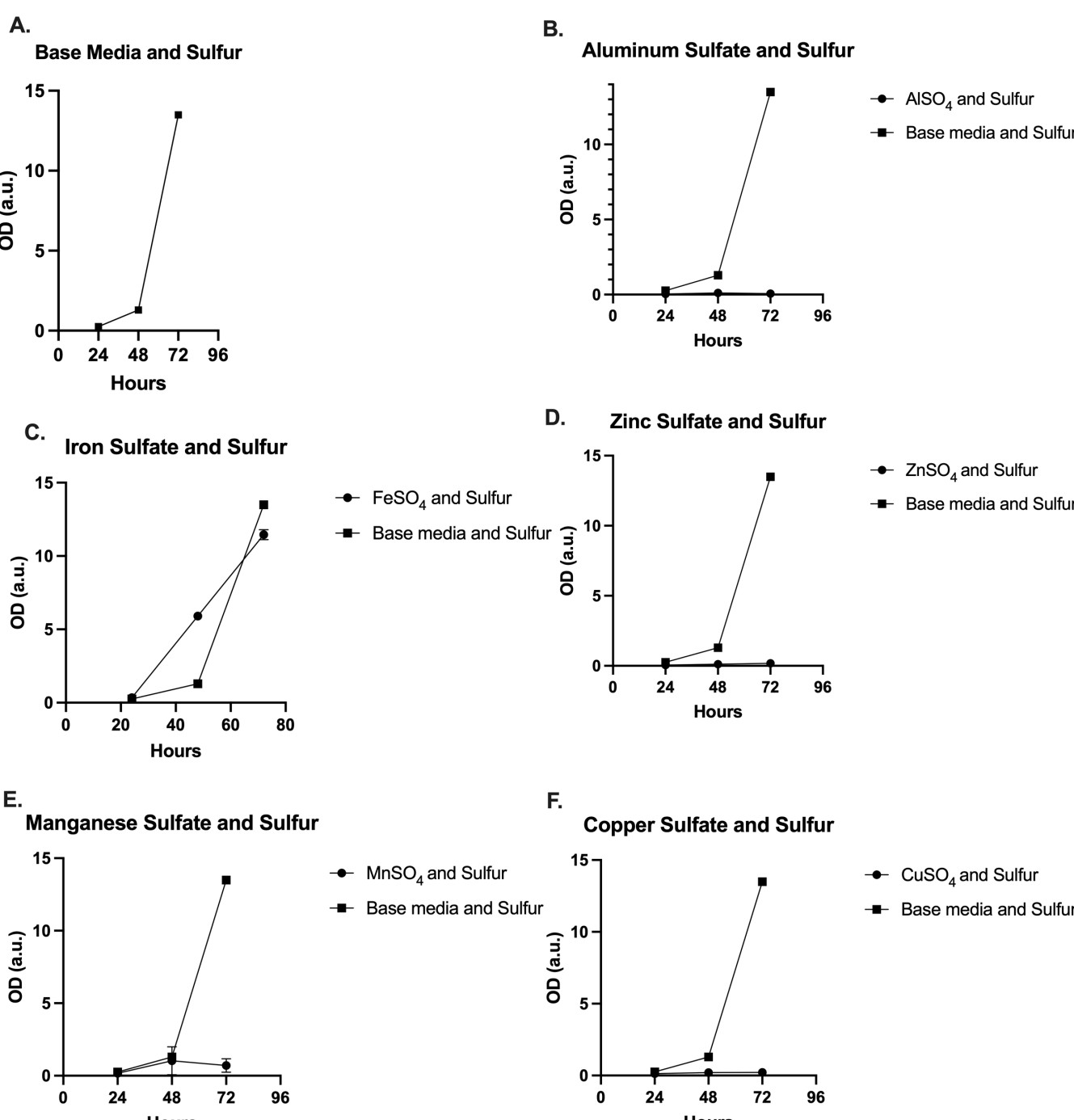

**FIG 4** Base media with elemental S and added solutes/metals AlSO$_4$, FeSO$_4$, ZnSO$_4$, MnSO$_4$, and CuSO$_4$. (A) *H. anomala* growth in base media with elemental S. (B) AlSO$_4$ in base media with added elemental S had negligible growth. (C) FeSO$_4$ suppressed growth in base media, but had significant growth with added elemental S. (D) Organism growth was negligible with added elemental S and ZnSO$_4$. (E) *H. anomala* growth with MnSO$_4$ showed some minimal growth at 48 h with the added elemental S that declined slightly at 72 h. (F) Similar to CuSO$_4$ in base media, added elemental S did not rescue *H. anomala* growth.

triplicate testing (Fig. 3D). While high sodium had no effect on *H. anomala* in the base media, when elemental S was added, the organism could no longer tolerate the high sodium, which may indicate a branching pathway for survival. Again, the organism was not killed by the high sodium and could be streaked out for colony growth (data not shown), so rather the high sodium may be inducing a dormant state.

With the addition of elemental S, growth could be rescued for $FeSO_4$ but not for $AlSO_4$, $CuSO_4$, or $ZnSO_4$ (Fig. 4). Although *H. anomala* grew very well with high concentrations of S alone in base media, three of the four suppressive metals continued to inhibit measurable growth. This may indicate the S-metabolic pathway for the organism includes a distinct concentration of the inhibitory and the supportive solute/metals for controlled growth.

## DISCUSSION

The specificity of the metal/solute growth observed is an interesting trait exhibited by this strain of yeast and grants an understanding into the adaptable tolerance of this organism in a superfund site. Some insight can be gained by the individual suppressive versus growth patterns as to the metabolism of this strain of *H. anomala*. It is interesting that some of these metals and solutes in particular combinations induce a dormancy in growth. In all cases, the organism survived changes in growth media reasserting its status as an extremophile. The growth patterns that have emerged from this work form the basis for future work on the metabolomics of this particular strain of *H. anomala* from the Berkeley Pit. More specifically, gene expression in metabolic pathways can be probed for expression coupled with enzyme detection assays to better understand the metal-specific growth pathways. This deeper understanding may lead to a solid basis to modify this organism as deemed important for the bioremediation of superfund sites such as the Berkeley Pit in Butte, Montana.

## ACKNOWLEDGMENTS

This research was funded by the National Institutes of Health (NIH) NIGMS RISE program R25-GM07632. Original organism was provided by Andrea Stierle Ph.D., Department of Biomedical and Pharmaceutical Sciences, University of Montana, Missoula.

## AUTHOR AFFILIATIONS

[1]Department of Life Sciences, Salish Kootenai College, Pablo, Montana, USA
[2]Energy Keepers Inc., Polson, Montana, USA
[3]Missoula County, District Attorney Justice Court Prosecution, Missoula, Montana, USA

## AUTHOR ORCIDs

Kyle Roessler  http://orcid.org/0009-0004-5021-1831
Christina L. Rush  http://orcid.org/0000-0001-9730-3080

## AUTHOR CONTRIBUTIONS

Kyle Roessler, Investigation | Mariah C. Friedlander, Investigation | Marthe Y. VanSickle, Investigation | Christina L. Rush, Writing – original draft

## ADDITIONAL FILES

The following material is available online.

### Supplemental Material

**Figure S1 legend (Spectrum00444-24-s0001.docx).** PCR amplification from genomic *H. anomala* DNA.
**Supplemental figure (Spectrum00444-24-s0002.tiff).** Fig. S1.
**Table S1 legend (Spectrum00444-24-s0003.docx).** Growth tests.
**Supplemental table (Spectrum00444-24-s0004.tiff).** Table S1.

Open Peer Review

**PEER REVIEW HISTORY (review-history.pdf).** An accounting of the reviewer comments and feedback.

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
