## [Reviewer comments · Microbiology Spectrum]

Microbiology Spectrum

Hanensula anomala isolated from the Berkeley Pit, Butte, MT, is a metal specific extremophile

Christina Rush, Kyle Roessler, Mariah Friedlander, and Marthe VanSickle

Corresponding Author(s): Christina Rush, Salish Kootenai College

Review Timeline:

Submission Date:	February 19, 2024
Editorial Decision:	June 2, 2024
Revision Received:	July 16, 2024
Accepted:	July 19, 2024

Editor: Erik Hom

Reviewer(s): The reviewers have opted to remain anonymous.

Transaction Report:

DOI: <https://doi.org/10.1128/spectrum.00444-24>

Re: Spectrum00444-24 (Hanensula anomala isolated from the Berkeley Pit, Butte, MT, is a metal specific extremophile)

Dear Dr. Christina Lynn Rush:

Thank you for the privilege of reviewing your work. Below you will find my comments, instructions from the Spectrum editorial office, and the reviewer comments.

I was having a hard time getting a second reviewer for your work but rather than delay any further (sorry it has taken so long!), I have looked over your manuscript myself. I concur with Reviewer #1's comments and also ask that you format your manuscript abstract in a form typical of Spectrum. In your response cover letter (please provide one that is not just the cover page of your manuscript), please summarize the changes you have made. Other details for resubmission are below.

Revision Guidelines

Sincerely,
Erik Hom
Editor
Microbiology Spectrum

Reviewer #1 (Comments for the Author):

Ref: Spectrum00444-24

Manuscript Title: Hanensula anomala isolated from the Berkeley Pit, Butte, MT, is a metal specific extremophile.

General Comment:

The present work is a research report on the extremophilic bacteria isolated from Berkeley Pit, Butte. The manuscript is critically evaluated to detect errors in the writing of English. This manuscript is written well in the English language and enough to justify the findings reported in the study. Therefore, I will be glad to recommend the publication. However, before publication, I am here in some suggestions. A more detailed suggestions can be found in the specific comments below. I ask that the authors specifically address each of my comments in their response.

Specific Comments:

1. Add a short background on the topic (extremophiles) at the starting of the abstract.
2. Conclusion is missing from the Abstract. It should be added.
3. The results section require refinement in some subsections.
4. After going through the whole manuscript, I found few typos, grammatical mistakes, and problem related to framing of new sentences. Please correct and revise.
5. All the cited references must be critically checked where the volume number, issue number, and page numbers are missing, should be mentioned.
6. The manuscript is recommended for minor revision.

Metal specific extremophile

***Hanensula anomala* isolated from the Berkeley Pit, Butte, MT, is a metal specific
extremophile**

Kyle Roessler¹, Mariah C. Friedlander², Marthe Y. VanSickle³, Christina L. Rush¹

*¹Department of Life Sciences, Salish Kootenai College, 58138 U.S. Hwy 93 Pablo, MT
59855*

²Energy Keepers Inc. 43069 Kerr Dam Road, Polson, MT 59860

*³Missoula County, District Attorney Justice Court Prosecution, 200 West Broadway,
Missoula, MT 59802*

Correspondence

E-mail: kyle_roessler@skc.edu, christina_rush@skc.edu, info@energykeepers.org,
clerkofcourt@missoulacounty.us

Abstract

A yeast-like extremophile organism, *Hansenula anomala* has been isolated from the superfund site the Berkeley Pit Lake in Butte, Montana. Growth studies demonstrate *H. anomala* growth in some of the known Berkeley Pit Lake solutes. Microbial growth dynamics under controlled conditions were compared and related for *H. anomala* for multiple metal concentrations. Each solute/metal was tested separately at previously reported concentrations on the geochemistry of the Berkeley Pit lake in the first 0.2 m in spring (pH 2.5). *H. anomala* grew well with Sulfur, MgSO₄, CaSO₄, KCl and NaSO₄ and was inhibited with FeSO₄, MnSO₄, CuSO₄, AlSO₄, or ZnSO₄. With the addition of elemental sulfur, growth was observed for FeSO₄ indicating minimal growth rescue. PCR amplification of genomic DNA from the organism using known ribosomal primers indicate the strain to be ATCC8168 (CBS 5759).

Importance: Laboratory growth studies of a strain of *H. anomala* from the Berkeley Pit have found the organism to be metal specific indicating some unique metabolism possibilities. These studies show that this strain is metal dependent and provides information about the adaptable tolerance of organisms in superfund sites as well as giving a basis for future bioremediation development utilizing *H. anomala*.

Keywords

Bioremediation, Yeasts, Water Quality, Metabolism, Identification

Background

Hansenula anomala, (*Pichia anomala*, *Wickerhamomyces anomalus*) is a yeast-like organism that has been isolated from a number of harsh environments including anaerobic feed storage units, the intestine of *Anopheles stephensi* (a malaria vector), and low pH metal toxic waste water from a textile plant in Argentina (Olstorpe *et al.* 2010; Ricci *et al.* 2010; Fernández *et al.* 2012). The taxonomy of *H. anomala* varies with some phylogenetic studies placing the organism to species within the genus *Hansenula* (Kurtzman *et al.* 2003). Another genus, *Wickerhamomyces* is also found in the literature from separate taxonomic grouping however, for simplicity, this research report will use *H. anomala* as the organism's name (Kurtzman *et al.* 2003; Kurtzman 2010).

Within the various strains of *H. anomala*, many show antimicrobial activity and *H. anomala* has been shown to inhibit gram-negative bacteria including *Enterobacteriaceae* species and filamentous fungi including *Aspergillus*, *Botrytis* and *Penicillium* species (Polonelli *et al.* 1986; Passoth *et al.* 2003; Friel *et al.* 2007; Olstorpe *et al.* 2010). The mechanism for inhibition may be due to “killer proteins” described to consist of a variety of molecular masses, and where the variation in size may be attributed to various levels of glycosylation (Polonelli *et al.* 1986; Vustin *et al.* 1989; De Ingeniis *et al.* 2009). One of the “killer proteins,” a novel exo- β -1,3-glucanase, was isolated from the *H. anomala* strain NCYC 434 in 2011 and revealed activity attributed to the inhibition of fungal colonization on fruit including apples and citrus thus suggesting an attractive possibility for *H. anomala* to act as a biocontrol agent (Friel *et al.* 2005; Passoth *et al.* 2006; Makovitzki *et al.* 2007; Izgu *et al.* 2011).

Phytase activity is an additional characteristic in a number of *H. anomala* strains from which a phytase gene was sequenced to utilize it as a biotechnological tool in

removing phosphorous from animal waste to reduce the overall environmental footprint of farms (Olstorpe *et al.* 2009; Kaur *et al.* 2010; Vohra *et al.* 2010).

H. anomala has also demonstrated promise in the bioremediation of high concentration metal containing wastewater. Specifically, a chromate-resistant strain, M10 was shown to remove toxic chromium (Cr^{6+}) in batch cultures using cell-free extracts suggesting the potential for restoration of left-over waste mining sites (Fernández *et al.* 2012).

The geochemistry of the Berkeley Pit includes a high concentration of metals and organic compounds, which vary by depth, increasing dramatically in concentration for metals, specifically iron and sulfate. (Davis *et al.* 1988; Pellicori *et al.* 2005; Gammons *et al.* 2003). In addition, metal concentrations increase seasonally recording a higher measured content in almost all for the fall as opposed to the spring, possibly due to rain and snow melt creating a somewhat dilution effect in the spring versus the fall after a hot and dry summer (Davis *et al.* 1988; Pellicori *et al.* 2005; Gamons *et al.* 2003).

Due to the unique environment from which our isolate was found, we hypothesized that there would be interesting growth patterns. Here we show solute and metal repressive and/or suppressive growth from the Berkeley Pit Lake isolate, *H. anomala* strain ATCC8168 (CBS 5759) providing experimental data on the adaptable tolerance of this organism in superfund sites.

Materials and methods

Liquid media preparation and growth of H. anomala under controlled conditions

Liquid media was prepared at a pH of 2.5 using sulfuric acid, 6 g of potato dextrose, and varying combinations of metals. For each media preparation, 6 g of potato dextrose was

added to 100 mL of MilliQ water and autoclaved for sterility. Sulfuric acid was added drop-wise to cooled media in a sterile hood for a final pH of 2.5 at room temperature. Metals were dissolved in 100 ml MilliQ water and sterile filtered or autoclaved depending on metal solubility. The metal preparation was then added to room temperature potato dextrose pH 2.5 sulfuric acid media. The final volume of each prepared media was adjusted to 250 ml using MilliQ water. Media was stored at 4 °C for no longer than 8 days or used immediately. All of the metals were sulfate salts except for potassium chloride (KCl) and sulfur (S) to mimic the extremely high concentration of sulfate (SO₄) in the Berkeley Pit lake. KCl salt was used because both K and Cl are very close in concentration in the lake. S was also tested due to the high level of sulfur released in the lake.

H. anomala was first cultured onto potato dextrose agar plates pH 2.5 from previously isolated samples generously given from Andrea Stierle Ph.D. of the University of Montana in 2011. The plates were streaked for colony isolation and incubated at 30 °C for 72 h. All samples and plates were kept under sterile hood conditions for streaking to prevent contamination.

Single isolated colonies of *H. anomala* were inoculated into the prepared liquid media of potato dextrose pH 2.5 sulfuric acid from the potato dextrose agar plate pH 2.5 using a sterile, disposable inoculation loop into 1.5 ml of the media in the absence of metal/solute and incubated at 30°C, overnight at 180 RPM. The purpose of growth in the absence of any metal was to promote confluent growth in liquid culture before inoculation into the metal/solute-containing liquid media for study. The 1.5 ml inoculates were then added to 50 ml of metal containing media and grown overnight at 30 °C at a

180 RPM. All inoculations and *H. anomala* culturing were carried out in a sterile fume hood.

50 mL inoculations were incubated for 72 h at 30 °C at a 180 RPM and sampled each day at the same time by removing the flasks from the shaking incubator for sampling from a sterile fume hood. 1 ml was removed using sterile pipettes from each flask. The flasks were then returned to continue growth in the shaking incubator. The 1 mL sample was then measured for growth and confluence using Optical Density (OD) measurements against un-inoculated liquid media.

The OD measurements were taken three times a week, over a 72 h growth period to calculate growth. By 24 h, samples were confluent and serial dilutions with un-inoculated media were carried out. OD measurements were then calculated for cell growth with the multiplication of the dilution for each measurement. For replication, all growth measurements were conducted three times for each metal/solute liquid media. Growth data was calculated using all three measurements for each meta/solute for data correlation. Statistical analysis of growth data was thus performed from triplicate samples. After 72 h sampling and OD measurements samples were centrifuged and pellets frozen for future protein profile characterization.

Genomic DNA extraction and PCR analysis

For genomic DNA extraction, *H. anomala* was grown in potato dextrose liquid media pH 2.5 at 30°C for 24 h to confluency in a 1 ml preparation at an rpm of 150. Cells were pelleted via centrifugation and genomic DNA was isolated from *H. anomala* using the QIAGEN Dneasy blood & tissue kit 250 and the accompanying protocol ‘Purification of total DNA from yeast using the Dneasy blood & tissue kit (DY13 Aug 06).’

PCR amplification was carried out with primers: ITS3 GCATCGATGAACGCAGC and ITS4 TCCTCCGCTTAATTGATATGC, NS1 GTAGTCATATGCTTGTCTC and NS-8 TCCGCAGGTTCACCTACGGA, ITS1 TCCGTAGGTGAACCTGCGG and NS-8A CCTTCCGCAGGTTCACCTACGGAAACC, NL-1 GCATATCAATAAGCGGAGGAAAG and NL-4 GGTCCGTGTTTCAAGACGG, NL-3A GAGACCGATAGCGAA-CAAG and NL-7AR CCGACTTCCATGGCCACCGTCC, NL-E27 GGTAGCCAATGCCTCGTCA and NL-11R CCTTGTCCGTACCAGTTCTAAGT, NLG19A GGGAACGTGAGCTGGGTTTAGACCG and NL-13R GCGTTATCGTTTAAACAGATGTGCCG (Chen *et al.* 2000; HerasVazques *et al.* 2002; Kurtzman *et al.* 2003) and *Taq* polymerase (1X standard *Taq* reaction buffer pack, New England Biolabs) at standard conditions: for a 25 µl reaction, 2.5 µl 10X standard *Taq* reaction buffer, 0.5 µl 10 mM dNTPs, 1 µl 10 µl FWD primer, 1 µl 10 µl REV primer, ~50 ng genomic DNA (varied from extraction), 0.5 µl *Taq* DNA polymerase, QS to 25 µl with nuclease-free water. Thermocycling conditions were set up for ribosomal and mitochondrial gene amplification from genomic DNA with an initial denaturation step for 1 min at 95 °C followed by 35 cycles (30 s at 95 °C, 60 s at 48 °C, 2 min 68 °C,) and a final extension 72 °C 5 min. PCR amplifications were analyzed via 1 % molecular grade agarose with SYBR® safe DNA gel stain.

Results and Discussion

PCR strain analysis of H. anomala isolated from the Berkeley Pit

PCR analysis was carried out on genomic DNA to determine the strain of *H. anomala* used in the growth study (originally isolated from the Berkeley Pit Lake).

Primers were designed from previous papers sequencing *H. anomala* strains (Chen *et al.* 2000; HerasVazques *et al.* 2002; Kurtzman *et al.* 2003). PCR analysis showed amplification of bands for primers described in the methods section and show the strain to be ATCC8168 (CBS 5759).

H. anomala minimal growth requirements

Initial growth studies in minimal media were conducted varying additives including; tryptophan, lysine, arginine, valine, proline, glucose, skim milk, lactose, fructose and cellulose (Supplementary data, table 1). A summary of these initial results is that *H. anomala* ATCC8168 could grow in less acidic pH up to 6.5. The organism does not appear have any essential amino acids as it has average growth in minimal base media however there is suppressed growth in the presence of skim milk (powdered), and when lactose or cellulose were added. M9 salts with added metals and glucose also show no growth. Finally, the organism was shown to be catalase positive and urease positive (Supplementary data, table 1.)

Metal specific growth studies

H. anomala ATCC8168 (CBS 5759) growth with specific metals/solutes was measured in a liquid media of nutrient potato dextrose broth pH 2.5 with sulfuric acid and showed significant differences in growth patterns for different metal/solute conditions over the course of 24 h - 72 h growth periods. The laboratory metal/solute concentrations used were consistent with the solute concentrations measured by Pellicori *et al.* (2005) for the Berkeley Pit Lake in spring at a depth of 0.2 m (Table 1). Fe(II) was not measured because its concentration at 0.2 m depth is negligible compared with Fe(III). The pH of the Berkeley Pit is acidic with a fifteen-year mean pH of 2.63 (Gammons *et al.* 2003).

H. anomala does not require metals for growth, as it grows well in base media and with added elemental sulfur at a concentration of 4530 mg/L (Pellicori *et al.* 2005), suggesting the organism has most likely evolved to utilize sulfur in metabolism pathways rather than simply tolerating it (Table 1). Additionally, the organism also showed significant growth in a base media of potato dextrose broth at a pH of 2.5 with sulfuric acid. Individual metals were added to the base media (Figure 1) (a) magnesium (302 mg/L), (b) calcium (420 mg/L) (c) potassium (14 mg/L) and (d) iron (263 mg/L). Each of these individually added metal showed significant growth as expected.

Interestingly, although all the tested metals and solutes were derived from those measured by Pellicori *et al.*, (2005), five metals suppressed growth when added to the base media. Figure 2 shows negligible growth in base media with (b) FeSO₄, (c) MnSO₄, (d) CuSO₄, (e) AlSO₄, or (f) ZnSO₄. To examine whether the metals iron, manganese, copper, aluminum or zinc were actually killing the organism or simply inhibiting growth, inoculates from the 72 h liquid media were streaked out on solid base media and incubated for 24 h at 30°C. Robust colony growth was observed indicating an inhibition of growth in the metal containing media, not cell death (results not shown).

To further probe the nature of the inhibition, liquid base media with the suppressive metals was inoculated and then provided with additional elemental sulfur. The hypothesis of the inhibition process was that added elemental sulfur could reverse the suppressive effects of the metals in question. As a control, the initial metals/solutes that resulted in significant organism growth in base media were tested with elemental sulfur (Figure 3.) Surprisingly, NaSO₄ in base media with elemental sulfur had negligible growth even over triplicate testing (Fig. 3.d.). While high sodium had no effect on *H.*

anomala in the base media, when elemental sulfur was added, the organism could no longer tolerate the high sodium, which may indicate a branching pathway for survival. Again, the organism was not killed by the high sodium and could be streaked out for colony growth (data not shown), so rather the high sodium may be inducing a dormant growth state.

With the addition of elemental sulfur, growth could be rescued for FeSO₄ but not for AlSO₄, CuSO₄ or ZnSO₄ (Fig 4). Although *H. anomala* grew very well with high concentrations of sulfur alone in base media, three of the four suppressive metals continued to inhibit measurable growth. This may indicate the sulfur-metabolic pathway for the organism includes a distinct concentration of the inhibitory and the supportive solute/metals for controlled growth.

The specificity of the metal/solute growth observed is an interesting trait exhibited by this strain of yeast and grants an understanding into the adaptable tolerance of this organism in a superfund site. Some insight can be gained by the individual suppressive versus growth patterns as to the metabolism of this strain of *H. anomala*. It is interesting that some of these metals and solutes in particular combinations induce a dormancy in growth. In all cases the organism survived changes in growth media reasserting its status as an extremophile. The growth patterns that have emerged from this work form the basis for future work on the metabolomics of this particular strain of *H. anomala* from the Berkeley Pit. More specifically, gene expression in metabolic pathways can be probed for expression coupled with enzyme detection assays to better understand the metal specific growth pathways. This deeper understanding may lead to a

solid basis to modify this organism as deemed important for bioremediation of superfund sites such as the Berkeley Pit in Butte, Montana.

Acknowledgements

This research was funded by the National Institutes of Health (NIH) NIGMS RISE program R25-GM07632. Original organism was provided by Andrea Stierle Ph.D., Department of Biomedical and Pharmaceutical Sciences, University of Montana, Missoula.

Conflict of Interest

The authors of this paper certify that they have NO affiliations with or involvement in any organization or entity with a financial interest (such as patent or stock ownership, membership of a company board of directors, membership of an advisory board or committee for a company, or consultancy for or receipt of speaker's fees from a company) in the subject matter or materials discussed in this manuscript.

References

- Davis A., A.D., The aqueous geochemistry of the Berkeley Pit, Butte, Montana, U.S.A. Applied Geochemistry, 1988. 4: p. 23-36.
- De Ingeniis J., R.N., Ciana M., Mannazzu I., *Pichia anomala* DBVPG 3003 secretes a ubiquitin-like protein that has antimicrobial activity. Appl Environ Microbiol, 2009. 75: p. 1129-1134.
- Fernandez, P.M., et al., Removal efficiency of Cr⁶⁺ by indigenous *Pichia* sp. isolated from textile factory effluent. ScientificWorldJournal. 2012: p. 708213.
- Friel, D., M. Vandenbol, and M.H. Jijakli, Genetic characterization of the yeast *Pichia anomala* (strain K), an antagonist of postharvest diseases of apple. J Appl Microbiol, 2005. 98(3): p. 783-8.
- Gammons CH., W.S., Jonas JP., Madison JP., Geochemistry of the rare-earth elements and uranium in the acidic Berkeley Pit lake, Butte, Montana. Chemical Geology, 2003. 198: p. 269-288.
- Izgu, D.A., R.A. Kepekci, and F. Izgu, Inhibition of *penicillium digitatum* and *penicillium italicum* in vitro and in planta with Panomycocin, a novel exo-beta-1,3-glucanase isolated from *Pichia anomala* NCYC 434. Antonie Van Leeuwenhoek. 99(1): p. 85-91.

Kaur, P., et al., Pphy--a cell-bound phytase from the yeast *Pichia anomala*: molecular cloning of the gene PPHY and characterization of the recombinant enzyme. *J Biotechnol.* 149(1-2): p. 8-15.

Kurtzman, C.P., Phylogeny of the ascomycetous yeasts and the renaming of *Pichia anomala* to *Wickerhamomyces anomalus*. *Antonie Van Leeuwenhoek.* 99(1): p. 13-23.

Kurtzman, C.P., C.J. Robnett, and E. Basehoar-Powers, Phylogenetic relationships among species of *Pichia*, *Issatchenkia* and *Williopsis* determined from multigene sequence analysis, and the proposal of *Barnettozyma* gen. nov., *Lindnera* gen. nov. and *Wickerhamomyces* gen. nov. *FEMS Yeast Res*, 2008. 8(6): p. 939-54.

Makovitzki, A., et al., Inhibition of fungal and bacterial plant pathogens in vitro and in planta with ultrashort cationic lipopeptides. *Appl Environ Microbiol*, 2007. 73(20): p. 6629-36.

Olstorpe, M. and V. Passoth, *Pichia anomala* in grain biopreservation. *Antonie Van Leeuwenhoek.* 99(1): p. 57-62.

Olstorpe, M., J. Schnurer, and V. Passoth, Screening of yeast strains for phytase activity. *FEMS Yeast Res*, 2009. 9(3): p. 478-88.

Passoth, V., et al., Biotechnology, physiology and genetics of the yeast *Pichia anomala*. FEMS Yeast Res, 2006. 6(1): p. 3-13.

Pellicori DA., G.C., Poulson SR., Geochemistry and stable isotope composition of the Berkeley pit lake and surrounding mine waters, Butte, Montana. Applied Geochemistry, 2005. 20: p. 2116-2137.

Polonelli, L., et al., Potential therapeutic effect of yeast killer toxin. Mycopathologia, 1986. 96(2): p. 103-7.

Ricci, I., et al., The yeast *Wickerhamomyces anomalus* (*Pichia anomala*) inhabits the midgut and reproductive system of the Asian malaria vector *Anopheles stephensi*. Environ Microbiol. 13(4): p. 911-21.

Vohra, A., P. Kaur, and T. Satyanarayana, Production, characteristics and applications of the cell-bound phytase of *Pichia anomala*. Antonie Van Leeuwenhoek. 99(1): p. 51-5.

Vustin, M.M., et al., [Killer protein, formed by the yeast *Hansenula Anomala* (Hansen) H. et P. Sydow]. Dokl Akad Nauk SSSR, 1989. 308(5): p. 1251-5.

Table 1 Solute concentrations for Berkeley pit lake, Spring 2003 by Pellicori 2005 at a depth of 0.2 m and liquid

media preparation solute concentrations

	Al	Ca	Cu	Fe III	Mg	Mn	K	Na	Zn	SO ₄
Berkeley pit lake	134	420	64.8	263	302	161	14	70	275	4530
	AlSO ₄	CaSO ₄	CuSO ₄	FeSO ₄	MgSO ₄	MnSO ₄	KCl	NaSO ₄	ZnSO ₄	Sulfur
Liquid media	134	420	64	263	300	161	19	70	275	4.53

* all concentrations are in mg/L

Figure 1. Base media added solutes/metals KCl, MgSO₄, CaSO₄ and NaSO₄. **a.** *H. anomala* grew at nearly the same rate as no added metal in the presence of KCl. **b.** Growth of the organism was slowed by approximately half with the addition of MgSO₄. **c.** CaSO₄ was within the error rate for matching growth with no added metal. **d.** NaSO₄ appeared to increase *H. anomala* growth in relationship to no added solute (metal).

Figure 2. Base media added solutes/metals AlSO₄, FeSO₄, ZnSO₄, MnSO₄ and CuSO₄. **a.** *H. anomala* growth in base media. **b.** No observed growth with added AlSO₄. **c.** *H. anomala* exhibited no growth when FeSO₄ was added to base media. **d.** No measurable growth was recorded with the addition of ZnSO₄ to base media. **e.** Very little measurable growth was observed with the addition of MnSO₄ to base media at 72 h post inoculation. **f.** Some growth was recorded for CuSO₄ additive media at 72 h post inoculation.

a) Potassium Chloride and Sulfur

b) Magnesium Sulfate and Sulfur

c) Calcium Sulfate and Sulfur

d) Sodium Sulfate and Sulfur

Figure 3. Base media with elemental Sulfur and added solutes/metals KCl, MgSO₄, CaSO₄ and NaSO₄. **a.** *H. anomala* had increased growth with KCl and elemental Sulfur than without Sulfur. **b. and c.** Organism growth was nearly as high as base media and Sulfur with added MgSO₄ and CaSO₄. **d.** Growth with NaSO₄ and elemental Sulfur was suppressed.

Figure 4. Base media with elemental Sulfur and added solutes/metals AlSO₄, FeSO₄, ZnSO₄, MnSO₄, and CuSO₄. **a.** *H. anomala* growth in base media with elemental Sulfur. **b.** AlSO₄ in base media with added elemental sulfur had negligible growth. **c.** FeSO₄ suppressed growth in base media, but had significant growth with added elemental sulfur. **d.** Organism growth was negligible with added elemental sulfur and ZnSO₄. **e.** *H. anomala* growth with MnSO₄ showed some minimal growth at 48 h with the added elemental sulfur that declined slightly at 72 h. **f.** Similar to CuSO₄ in base media, added elemental sulfur did not rescue *H. anomala* growth.

Supplementary Table 1. Growth tests observed at 0.6 A.U. cutoff, supplements for each sample are either metal, solute, amino acid or variable pH with sulfuric acid

Media	Supplement	OD \geq 0.6 A.U. Confluent growth
Minimal media	Tryptophan	Yes
Minimal media	Lysine	Yes
Minimal media	Arginine	Yes
Minimal media	Valine	Yes
Minimal media	Proline	Yes
Minimal media	None	Yes
Minimal media	Glucose	Yes
Minimal media	Skim Milk	No
Minimal media	Lactose	No
Minimal media	Fructose	Yes
Minimal media	Cellulose	No
M9 Salts	Glucose	No
M9 Salts	MgSO ₄ , Glucose	No
M9 Salts	CaSO ₄ , Glucose	No
M9 Salts	MgSO ₄ , CaSO ₄ , Glucose	No
Potato Dextrose	pH 1	No
Potato Dextrose	pH 1.5	Yes
Potato Dextrose	pH 2	Yes
Potato Dextrose	pH 2.5	Yes
Potato Dextrose	pH 3	Yes
Potato Dextrose	pH 3.5	Yes
Potato Dextrose	pH 4	Yes
Potato Dextrose	pH 4.5	Yes
Potato Dextrose	pH 5	Yes
Potato Dextrose	pH 5.5	Yes
Potato Dextrose	pH 6	Yes
Potato Dextrose	pH 6.5	Yes

Supplementary Figure 1. PCR amplification from genomic *H. anomala* DNA.

Amplified bands from genomic DNA. Lane 2: ITS3, ITS4; Lane 3: NS-1, NS-8; Lane 4: NS-1, NS-8A; Lane 5: NL-1, NL-4; Lane 6: NL-3A, NL-7AR; Lane 10: NL-E27, NL-11R; Lane 11: NL-G19A, NL-13R.

Response to Reviewers

Reviewer #1 (Comments for the Author):

Ref: Spectrum00444-24

Manuscript Title: *Hanensula anomala* isolated from the Berkeley Pit, Butte, MT, is a metal specific extremophile.

General Comment:

The present work is a research report on the extremophilic bacteria isolated from Berkeley Pit, Butte. The manuscript is critically evaluated to detect errors in the writing of English. This manuscript is written well in the English language and enough to justify the findings reported in the study. Therefore, I will be glad to recommend the publication. However, before publication, I am here in some suggestions. A more detailed suggestions can be found in the specific comments below. I ask that the authors specifically address each of my comments in their response.

Specific Comments:

1. Add a short background on the topic (extremophiles) at the starting of the abstract.
2. Conclusion is missing from the Abstract. It should be added.
3. The results section require refinement in some subsections.
4. After going through the whole manuscript, I found few typos, grammatical mistakes, and problem related to framing of new sentences. Please correct and revise.
5. All the cited references must be critically checked where the volume number, issue number, and page numbers are missing, should be mentioned.
6. The manuscript is recommended for minor revision.

Changes to the manuscript are as follows:

- Conclusion sentence added to the Abstract
- Short background on extremophiles added to the manuscript
- Re-worded sections in the results for easier interpretation
- Typos and grammatical errors have been fixed as far as the author could find through the manuscript (please see track changes for details)
- Cited references have been reworked to ensure proper citation with all authors, proper page and volume numbers
- Citations have been changed to the style format of Microbiology Spectrum
- Paper sections have been changed to the style format of Microbiology Spectrum
- Figures have been changed to TIFF formats
- Supplementary figures have been separated from legends for separate uploads

Re: Spectrum00444-24R1 (Hanensula anomala isolated from the Berkeley Pit, Butte, MT, is a metal specific extremophile)

Dear Dr. Christina Lynn Rush:

Your manuscript has been accepted, and I am forwarding it to the ASM production staff for publication. Your paper will first be checked to make sure all elements meet the technical requirements. ASM staff will contact you if anything needs to be revised before copyediting and production can begin. Otherwise, you will be notified when your proofs are ready to be viewed.

Sincerely,
Erik Hom
Editor
Microbiology Spectrum